# Vaccine Vigilance System: Considerations on the Effectiveness of Vigilance Data Use in COVID-19 Vaccination

**DOI:** 10.3390/vaccines10122115

**Published:** 2022-12-10

**Authors:** Diana Araja, Angelika Krumina, Zaiga Nora-Krukle, Uldis Berkis, Modra Murovska

**Affiliations:** 1Institute of Microbiology and Virology, Riga Stradins University, 5 Ratsupites Str., LV-1067 Riga, Latvia; 2Department of Infectology, Riga Stradins University, 3 Linezera Str., LV-1006 Riga, Latvia; 3Latvian Centre of Infectious Diseases, Riga East University Hospital, 3 Linezera Str., LV-1006 Riga, Latvia; 4Development and Project Department, Riga Stradins University, 16 Dzirciema Str., LV-1007 Riga, Latvia

**Keywords:** pharmacovigilance, EudraVigilance, COVID-19 vaccines, adverse drug reactions (ADRs), chronic fatigue syndrome, neurological and autoimmune diseases

## Abstract

(1) Background: The safety of medicines has been receiving increased attention to ensure that the risks of taking medicines do not outweigh the benefits. This is the reason why, over several decades, the pharmacovigilance system has been developed. The post-authorization pharmacovigilance system is based on reports from healthcare professionals and patients on observed adverse reactions. The reports are collected in databases and progressively evaluated. However, there are emerging concerns about the effectiveness of the established passive pharmacovigilance system in accelerating circumstances, such as the COVID-19 pandemic, when billions of doses of new vaccines were administered without a long history of use. Currently, health professionals receive fragmented new information on the safety of medicines from competent authorities after a lengthy evaluation process. Simultaneously, in the context of accelerated mass vaccination, health professionals need to have access to operational information—at least on organ systems at higher risk. Therefore, the aim of this study was to perform a primary data analysis of publicly available data on suspected COVID-19 vaccine-related adverse reactions in Europe, in order to identify the predominant groups of reported medical conditions after vaccination and their association with vaccine groups, as well as to evaluate the data accessibility on specific syndromes. (2) Methods: To achieve the objectives, the data publicly available in the EudraVigilance European Database for Suspected Adverse Drug Reaction Reports were analyzed. The following tasks were defined to: (1) Identify the predominant groups of medical conditions mentioned in adverse reaction reports; (2) determine the relative frequency of reports within vaccine groups; (3) assess the feasibility of obtaining information on a possibly associated syndrome—myalgic encephalomyelitis/chronic fatigue syndrome (ME/CFS). (3) Results: The data obtained demonstrate that the predominant medical conditions induced after vaccination are relevant to the following categories: (1) “General disorders and administration site conditions”, (2) “nervous system disorders”, and (3) “musculoskeletal and connective tissue disorders”. There are more reports for mRNA vaccines, but the relative frequency of reports per dose administered, is lower for this group of vaccines. Information on ME/CFS was not available, but reports of “chronic fatigue syndrome” are included in the database and accessible for primary analysis. (4) Conclusions: The information obtained on the predominantly reported medical conditions and the relevant vaccine groups may be useful for health professionals, patients, researchers, and medicine manufacturers. Policymakers could benefit from reflecting on the design of an active pharmacovigilance model, making full use of modern information technologies, including big data analysis of social media and networks for the detection of primary signals and building an early warning system.

## 1. Introduction

The safety, quality, and effectiveness of medicinal products are the determining criteria for the marketing authorization and use in the treatment process. A system for monitoring the safety of medicines, including vaccines, has been gradually developed worldwide. The marketing authorization of a new medicinal product is granted on the basis of a favorable benefit–risk balance for its target population and indication. However, not all risks will have been identified at the time when an initial authorization is sought and many of the risks associated with the use of a medicine will only be discovered or fully characterized after authorization [1]. Therefore, post-authorization risk monitoring is important. Post-authorization surveillance relies mostly on the collection of reports from healthcare professionals and patients. The reports collected on suspected adverse reactions to medicines are progressively evaluated. On the basis of this evaluation, the competent authorities may decide to make changes to the safety information on the medicinal product.

International institutions, organizations, and national competent authorities are setting up various platforms for collecting adverse drug reaction (ADR) reports and competent committees for evaluating these reports. At the international level, the World Health Organization (WHO) Program for International Drug Monitoring is a global collaboration aiming at the development of vigilance practices worldwide. The Uppsala Monitoring Centre provides VigiBase Services as a WHO global database of reported ADRs [2].

In the United States of America (US), vaccine-related vigilance is fulfilled by the Vaccine Adverse Event Reporting System (VAERS). VAERS is co-monitored by the Food and Drug Administration (FDA) and the Centers for Disease Control and Prevention (CDC) [3]. VAERS aims to detect potential signals of ADRs that occur after the administration of vaccines licensed or authorized in the US. Reports are collected from all stakeholders: Healthcare providers, vaccine manufacturers, vaccine recipients (or their parents/guardians, and state immunization programs. Safety signals detected in VAERS are further evaluated in safety systems, such as the CDC Vaccine Safety Datalink (VSD) or the Clinical Immunization Safety Assessment (CISA) project [3].

In the European Union (EU), Directive 2001/83/EC of the European Parliament and of the Council of 6 November 2001 on the Community code related to medicinal products for human use [4], introduced the term “pharmacovigilance” into EU legislation and established the obligation of stakeholders to support the development of a pharmacovigilance system. Pharmacovigilance is defined as the science and activities related to the detection, assessment, understanding, and prevention of adverse effects or any other medicine-related problems [5]. Vaccine vigilance is the part of pharmacovigilance that deals with the safety of vaccine administration.

The European Medicines Agency (EMA) coordinates the EU pharmacovigilance system and provides processes to support pharmacovigilance. In particular, EudraVigilance is a data processing network and management system for the reporting and assessment of suspected ADRs . In post-authorization surveillance phase, EudraVigilance provides a module for Individual Case Safety Reports (ICSRs) submitted by healthcare professionals and patients [6]. EudraVigilance first operated in December 2001, with access to the database being governed by the EudraVigilance access policy. In November 2017, the new full functionalities of EudraVigilance were launched, including the extensive web access to data on suspected ADRs [7]. 

The established Pharmacovigilance Risk Assessment Committee (PRAC) evaluates safety signals identified in EudraVigilance and may recommend changes in medicinal product safety information [6]. The PRAC is responsible for assessing all aspects of the risk management of medicinal products, thus ensuring that medicines approved for the EU market are optimally used by maximizing their benefits and minimizing risks [8].

The issue of evidence regarding the safety of medicinal products has become particularly relevant in recent years due to the acceleration of the clinical research process and reductions in real research time, especially under the circumstances of the coronavirus disease caused by the SARS-CoV-2 virus (COVID-19) pandemic. Vaccines remain the best prophylactic action created by science and are expected to save millions of lives. Watson et al. [9] estimated that by mathematical modeling based on officially reported COVID-19 deaths, vaccinations prevented 14·4 million deaths from COVID-19 in 185 countries and territories between 8 December 2020 and 8 December 2021. This estimate rose to 19·8 million deaths from COVID-19 averted when researchers used excess deaths as an estimate of the true extent of the pandemic, during the first year of COVID-19 vaccination [9].

As a result, vaccine safety is of high importance and the role of the benefit-risk assessment process is increasing. In these circumstances, there are emerging concerns about the effectiveness of the established passive pharmacovigilance system, when billions of doses of new vaccines were administered without a significant experience of use. Currently, health professionals receive fragmented new information on the safety of medicines from competent authorities after a lengthy evaluation process. However, in the context of accelerated mass vaccination, health professionals need to have access to operational information—at least on organ systems at higher risk.

Therefore, the question arises as to whether vaccine vigilance system data are used effectively in medical practice and research. In 2022, a significant number of scientific articles have been published on suspected COVID-19 vaccine adverse reactions and vaccination complications. However, these publications are mostly based on case studies and subsequent literature reviews of these studies. These publications provide information to the target audience according to specialization, but do not fulfill a comprehensive scoping analysis. The scoping analysis should use data from the above-mentioned vigilance platforms and databases of competent institutions and organizations.

To make an overview of the use of the EudraVigilance database in research, a literature search on the PubMed database was performed. A total of 27 articles were selected by keywords: “EudraVigilance”, “COVID-19”, and “vaccine”. The information on the purposes of these studies is summarized in Table 1.

The data show that the use of EudraVigilance data in research is gradually increasing. EudraVigilance data are predominantly used to study specific adverse reactions and potentially related syndromes and diagnoses. A broader context was observed in some studies, such as the method of Cari et al. [19], whereby an analysis of European data was performed on cardiovascular, neurological, and pulmonary events following vaccination with COVID-19 vaccines. Moreover, the extended use of EudraVigilance data is seen in the Montano study [23], which aimed to provide a risk assessment of the adverse reactions related to the COVID-19 vaccines manufactured by AstraZeneca, Janssen, Moderna, and Pfizer-BioNTech that have been in use in the EU and US between December 2020 and October 2021. Reported COVID-19 vaccine exposures and adverse reactions were compared to the reported influenza vaccine exposures and adverse reactions between 2020 and 2021, in order to obtain comparative data on these vaccines [23].

However, there is a lack of studies investigating the comparative magnitudes of prevalent medical conditions reported after COVID-19 vaccination, as well as their relevance to specific vaccine groups.

Therefore, the aim of this study was to perform a primary data analysis of publicly available data on suspected COVID-19 vaccine-related adverse reactions in Europe, in order to identify the predominant groups of reported medical conditions after vaccination and their association with vaccine groups, as well as to evaluate the data accessibility on specific syndromes.

To achieve the objectives, the following tasks were defined:

Task 1—to identify the predominant groups of medical conditions mentioned in adverse reaction reports in EudraVigilance database;

Task 2—to determine the relative frequency of reports within vaccine groups;

Task 3—to assess the feasibility of obtaining information on a possibly associated syndrome (myalgic encephalomyelitis/chronic fatigue syndrome (ME/CFS)).

## 2. Materials and Methods

The ICSRs information submitted by healthcare professionals and patients in EudraVigilance database is encoded using the Medical Dictionary for Regulatory Activities (MedDRA) terms [37]. MedDRA represents a structured classification system of medical conditions, but not a classification of diagnoses according to the International Classification of Diseases (ICD). Therefore, MedDRA terminology and classification system are not widely used by healthcare professionals in daily work, and additional information is provided below. 

In the late 1990s, MedDRA was developed by the International Council for Harmonization of Technical Requirements for Pharmaceuticals for Human Use (ICH), in order to facilitate the international exchange of regulatory information on medical products for human use. The scope of MedDRA covers pharmaceuticals, biologics, vaccines, and combination products. MedDRA subscriptions are available free of charge to all regulators worldwide, while paid subscriptions are determined on a sliding scale linked to companies’ annual turnover. Academic institutions and healthcare providers can access MedDRA free of charge from the Maintenance and Support Services Organization (MSSO), but specific institutional permission is required [38].

Therefore, explanations of three categories of MedDRA medical conditions [39] that were identified as predominantly reported in ICSRs on COVID-19 vaccines are herein provided: general disorders and administration site conditions (administration site reactions; body temperature conditions; complications associated with device; fatal outcomes; general system disorders (not elsewhere classified (NEC)); therapeutic and nontherapeutic effects (excl. toxicity); tissue disorders NEC);nervous system disorders (central nervous system infections and inflammations; central nervous system vascular disorders; congenital and peripartum neurological conditions; cranial nerve disorders (excl. neoplasms); demyelinating disorders; encephalopathies; headaches; increased intracranial pressure and hydrocephalus; mental impairment disorders; movement disorders (incl. parkinsonism); nervous system neoplasms benign; nervous system neoplasms malignant and unspecified NEC; neurological disorders NEC; neurological disorders of the eye; neuromuscular disorders; peripheral neuropathies; seizures (incl. subtypes); sleep disturbances (incl. subtypes); spinal cord and nerve root disorders; structural brain disorders);musculoskeletal and connective tissue disorders (bone disorders (excl. congenital and fractures); connective tissue disorders (excl. congenital); fractures; joint disorders; muscle disorders; musculoskeletal and connective tissue deformities (incl. intervertebral disc disorders); musculoskeletal and connective tissue disorders congenital; musculoskeletal and connective tissue disorders NEC; musculoskeletal and connective tissue neoplasms; synovial and bursal disorders; tendon, ligament, and cartilage disorders).

The data were searched in the EudraVigilance European Database for Suspected Adverse Drug Reaction Reports (EDSADRR), as a module for ICSRs submitted by healthcare professionals and patients, on the official website http://www.adrreports.eu/ (accessed on 17 September 2022).

A quantitative analysis was performed during Task 1, for all COVID-19 vaccines included in EudraVigilance EDSADRR on 17 September 2022. Specifically for Task 2, data normalization using the total number of administered doses has been applied. The total number of administered doses was downloaded from the platform “Our World in Data” on 17 September 2022 (Our World in Data is a project of the Global Change Data Lab, a non-profit organization based in the United Kingdom (Registered Charity Number 1186433), produced as a collaborative effort with researchers at the University of Oxford, who are the scientific contributors to the website content [40]). In the scope of Task 3, a qualitative search was carried out on EDSADRR to identify reports of “myalgic encephalomyelitis/chronic fatigue syndrome” and “chronic fatigue syndrome”, potentially related to COVID-19 vaccines and to estimate the reporting rates of “chronic fatigue syndrome” after COVID-19 vaccination in the European population.

## 3. Results

Different stakeholders have different levels of access to EudraVigilance data, although the EDSADRR is a tool available for healthcare professionals, patients, and the general public, in accordance with the EMA’s EudraVigilance access policy [41]. The EMA warns that the information on the EDSADRR website concerns suspected side effects of COVID-19 vaccines, i.e., medical events that have been observed following their administration, but these are not necessarily related to or caused by vaccines. The EMA’s scientific assessment considers many other factors, such as a patient’s medical history, the frequency of the suspected adverse reaction in the vaccinated population compared with the frequency in the general population, and whether it is biologically plausible that a vaccine could have caused the event. Only a detailed assessment of all available data allows for robust conclusions to be drawn regarding the benefits and risks of COVID-19 vaccines [42].

The EDSADRR comprises seven main Tabs that provide the following information: (1) The total number of individual cases; (2) the number of individual cases received over the last 12 months by geographic origin; (3) the number of individual cases by EEA countries; (4) the number of individual cases by reaction groups; (5) the number of individual cases for a selected reaction group; (6) the number of individual cases for a selected reaction; (7) line listing of individual cases, including the ICSR form, based on the selection of filtering conditions [42]. For the purposes of Task 1 and Task 2 of this research, EDSADRR Tab 4 on the number of individual cases by reaction groups was used; and for Task 3, EDSADRR Tab 7 with line listing of individual cases was used, including the ICSR form, based on the selection of filtering conditions, in order to obtain information on ICSRs reported for the following EU-authorized COVID-19 vaccines.

Table 2 below provides information on COVID-19 vaccines, which were included in the EudraVigilance EDSADRR database. In addition, their administered doses in EU and the percentage of adverse effects reported until 17 September 2022, were included in Table 2.

The data for seven dominating reaction groups reported in ICSRs were investigated for each of these vaccines. Figure 1 illustrates the results of Task 1 and the five predominant groups of medical conditions reported in ICSRs that are common to all five vaccines.

The data obtained show that the three predominant medical conditions induced after vaccination are relevant to the following categories: (1) “General disorders and administration site conditions”, (2) “nervous system disorders”, and (3) “musculoskeletal and connective tissue disorders”. Figure 1 shows the total number of ICSRs in each medical condition group, divided by seriousness. In accordance with EU legislation, a serious adverse reaction is an adverse reaction that results in death, is life-threatening, requires inpatient hospitalization or prolongation of existing hospitalization, results in persistent or significant disability or incapacity, or is a congenital anomaly/birth defect [4]. Notably, in the category “infections and infestations”, the number of ICSRs on serious suspected adverse reactions is three times more than the non-serious suspected adverse reactions.

Two specific groups of the seven predominant medical conditions are different for different vaccines. For TOZINAMERAN, the groups are “reproductive system and breast disorders” (this is the only vaccine with “reproductive system and breast disorders” among the seven dominant categories of suspected adverse reactions) and “skin and subcutaneous tissue disorders”. For CX-024414 and CHADOX1 NCOV-19, the two additional groups are “respiratory, thoracic and mediastinal disorders” and “skin and subcutaneous tissue disorders”. For AD26.COV2.S, “respiratory, thoracic and mediastinal disorders” and “investigations” (such as “neurological, special senses and psychiatric investigations”). For NVX-COV2373, “skin and subcutaneous tissue disorders” and “cardiac disorders” (this is the only vaccine with “cardiac disorders” among the seven dominant categories of suspected adverse reactions, with a relatively high proportion of serious reactions in this group) [42].

Figure 2 shows the results of Task 2 on the relative frequency of ICSRs for each vaccine by data normalization using the total number of administered doses in EU in the period until 17 September 2022.

The data of the EudraVigilance EDSADRR demonstrate that there are more ICSRs for mRNA vaccines, but the relative frequency of ICSRs per dose administered is lower for this group of vaccines. The relative frequency of ICSRs for Vector and Subunit vaccines exceeds the mRNA vaccines in all five categories. The situation is most critical for CHADOX1 NCOV-19 vaccine in all categories.

In Task 3, ME/CFS was chosen to assess the feasibility of obtaining information on a possibly associated syndrome. However, information on ME/CFS was not available, although ICSRs on “chronic fatigue syndrome” are included in the EudraVigilance EDSADRR database and accessible for primary analysis. The authors previously investigated chronic fatigue syndrome in the scope of ME/CFS and its growth in light of the COVID-19 pandemic [44,45]. Some researchers have noted that the preliminary findings raise concerns regarding a possible future ME/CFS-like pandemic in SARS-CoV-2 survivors [46,47]. Based on previous studies on other infections, researchers assume that 10% of COVID-19 survivors could develop ME/CFS [48]. A recent publication on the prospective observational study of post-COVID-19 chronic fatigue syndrome demonstrated that 19 of 42 post-COVID-19 syndrome patients with persistent moderate to severe fatigue and exertion intolerance 6 months following COVID-19 fulfilled the 2003 Canadian Consensus Criteria for ME/CFS [49].

At the same time, CFS has been reported in the scientific literature not only as a consequence of COVID-19, but also as a response to COVID-19 vaccination. Loosen et al. described the study based on data from the Disease Analyzer database (IQVIA) on 531,468 individuals who received a total of 908,869 SARS-CoV-2 vaccinations in 827 general practices in Germany between April and September 2021. The total number of ADRs documented was 28,287 (3.1% of all vaccinations). Pain in the limb (24.3%), fatigue (21.0%), dizziness (17.9%), joint pain (15.7%), fever (9.5%), nausea (7.5%), and myalgia (6.4%) were the most common ADRs documented among the 12,575 vaccinations with definite ADRs [50]. Sriwastava et al. performed a literature review and case series analysis on the spectrum of neuroimaging findings in post-COVID-19 vaccination. Researchers emphasized that the global vaccination campaign has been effective in reducing morbidity and mortality of COVID-19 in vaccinated individuals; however, several adverse effects following immunization (AEFI) have been noted [51]. It was mentioned that typical ADRs following vaccination include pain, swelling, localized erythema over the injection site, fever, chills, fatigue, myalgia, muscle pain, vomiting, arthralgia, and lymphadenopathy [52,53,54,55]. Mild neurological symptoms including headache, dizziness, myalgia, muscle spasms, and paresthesia have also been reported. A small number of case reports have demonstrated serious post-vaccination neurological sequelae ranging from generalized seizures, Guillain Barre Syndrome (GBS), and transverse myelitis [56]. Other neurological manifestations, such as facial nerve palsy, acute disseminated encephalomyelitis, and stroke have been reported in the VAERS related to COVID-19 Pfizer-BioNTech, Moderna, and Johnson & Johnson’s COVID-19 vaccines [51]. 

The results of the authors’ performed search for “chronic fatigue syndrome” in the EudraVigilance EDSADRR database Tab 7—line listing of individual cases, including ICSR forms, based on the selection of filtering conditions—are summarized in Figure 3.

Vaccine NVX-COV2373 was not reported in ICSRs on CFS in 2021; therefore, no data are presented for 2021. Vaccine CHADOX1 NCOV-19 is not included in Figure 3 since its data far exceed the data proportionality with other vaccines. In 2021, 26 ICSRs were reported per 10 million CHADOX1 NCOV-19 doses administered; and in 2022, 21,667 ICSRs were submitted per 10 million doses administered. Despite the significant reduction in administered doses of all vaccines, except for NVX-COV2373, in 2022, the number of ICSRs on CFS has increased manifold. The total number of ICSRs on CFS identified for COVID-19 vaccines in EDSADRR doubled in 2022 compared to 2021 [42], although, the number of vaccine doses administered in the EU until 17 September 2022 comprised a fifth of the number of doses administered in 2021, according to Our World in Data [43]. These data may support the hypothesis that CFS is related to COVID-19 vaccination, given that 6 months of follow-up are used for diagnosis.

## 4. Discussion

The pharmacovigilance system has been built over decades with the aim of protecting the health and lives of patients through both passive vigilance and encouraging the preventative use of vigilance data, in order to ensure a personalized approach to the selection of medicines for each patient’s needs based on drug safety data. In 1998, it was estimated that ADRs could account for more than 100,000 deaths in the US each year, making them the fourth most common cause of death after heart disease (nearly 750,000 deaths), cancer (530,000), and stroke (150,000) [57]. In 2008, a meta-analysis of ADR studies reported over 180,000 deaths and over 1 million injuries from ADRs in the US [58]. In the same year, the European Commission reported that an estimated 197,000 deaths per year in the EU were caused by ADRs and that the total cost to society of ADRs in the EU was EUR 79 billion [59]. 

In recent years, the use of precise numbers has mostly been avoided due to the argument that a deep causality assessment should be performed. For this reason, various assessment bodies, such as the PRAC were established, but questions remain regarding the capacity needed for the assessment of millions of ADRs reported each year. The first 6 years of the EU signal management system resulted in 453 recommendations issued by the PRAC, of which more than half were for medicine labeling changes [60]. 

From the perspective of healthcare professionals who participate in reporting, medical practitioners’ “input” in the system (measured in millions of reports) receive “output” in a relatively long process of institutional assessment, surveillance, and (in rare cases) the preparation of recommendations for marketing authorization holders (MAHs), in order to make changes in the summary of product characteristics (SmPC) and package leaflets. A MAH may send a direct healthcare professional communication (DHPC) to healthcare professionals to inform them of important new safety information about a medicinal product, but this is not mandatory. Some information is available on the EMA’s and national competent authorities’ websites if healthcare professionals proactively search on their own time. 

Vaccines retain their leading role as primary prevention for many infections. Simultaneously, it should be noted that increased vaccine precaution was also present before the COVID-19 pandemic. As shown by the literature review published in 2018, vaccines were the dominant category in publications on suspected ADRs [7]. Therefore, vaccines require the most active–vaccine vigilance model. The current vigilance approach could be suitable prior to the era of “fast-tracked COVID-19 vaccines”, but under “fast-tracked vaccination” circumstances, healthcare professionals need more effective measures to prevent avoidable health damage to patients.

It is therefore now justified to use modern methods such as machine learning and big data analysis of ADR databases and social media resources. As mentioned by the FDA, this type of active surveillance involves proactively obtaining and rapidly analyzing information that occurs in millions of individuals and is recorded in large healthcare data systems, in order to verify the safety signals identified through passive surveillance or to detect additional safety signals that may not have been reported as adverse events to passive surveillance systems [61]. The methodology for working with social media data has been in development by researchers since at least 2015 and has been sufficiently described in scientific literature, demonstrating proven, real-world data. This approach is intended to be more useful for healthcare professionals than the use of individual raw media data. 

In the scope of this research, the screening of possibilities to use EudraVigilance data revealed the potential for obtaining overall information, performing categorizations, and selecting outcomes for specific suspected adverse reactions, such as “chronic fatigue syndrome”. The results demonstrated that the “general disorders and administration site conditions” group significantly exceeds other categories of medical conditions following COVID-19 vaccination. However, the “general disorders and administration site conditions” group included a broad spectrum of conditions, from “nontherapeutic effects” to “fatal outcomes”; therefore, a separate investigation should be carried out to investigate all these data. Noticeably, the group “chronic fatigue syndrome” also belongs to the “general disorders and administration site conditions” group according to MedDRA.

Two other dominant reaction groups for COVID-19 vaccination are “nervous system disorders” and “musculoskeletal and connective tissue disorders”, which also have a vast range of medical conditions. Recent publications of 2022 demonstrate relapses of previous autoimmune diseases or the development of new autoimmune or autoinflammatory conditions following vaccination, proven by literature reviews and case studies [62,63,64,65,66]. New-onset autoimmune phenomena after COVID-19 vaccination have been reported increasingly (e.g., immune thrombotic thrombocytopenia, autoimmune liver diseases, Guillain-Barré syndrome, IgA nephropathy, rheumatoid arthritis, and systemic lupus erythematosus) [63]. Many of these autoimmune syndromes meet sufficient criteria for the diagnosis of Adjuvant-Induced Autoimmune Syndrome (ASIA syndrome) [64]. As mentioned by researchers, vaccination is one of the most effective interventions to substantially reduce severe disease and death due to SARS-CoV-2 infection. Consequently, vaccination programs were being rolled out globally, but most of these vaccines have been approved without extensive studies on their side effects and efficacy [63]. Therefore, autoimmune manifestations that may lead to autoimmune diseases need additional investigations, as there are assumptions that their real score is more influential.

Moreover, it should be considered that all vaccines in the EudraVigilance database have the group “infections and infestations” listed as one of the seven predominant reaction groups after vaccination. In this context, the authors of the present article note their previous study on COVID-19 pandemic-revealed consistencies and inconsistencies in healthcare, which used statistical data to indicate that completely vaccinated persons tended to be infected with COVID-19 more often than unvaccinated persons in the first 4 months of 2022 in Latvia [67]. This issue highlights another dimension of pharmacovigilance—the effectiveness of the medicinal product, which, in the case of vaccines, can also manifest itself as “vaccine resistance”. Furthermore, one more causality that might be worth investigating is related to the ensuring of infection-preventive measures after the administration of a vaccine. Questions arise on whether the person is warned about the need to avoid widely visited places for a few days, whether paid vacation days are offered, and whether medical supervision takes place. Therefore, ensuring infection-preventive measures after the administration of a vaccine hypothetically remains topical.

Returning to the research question of the current research, it can be highlighted that a formal passive pharmacovigilance system is operational and vigilance data are partly publicly available. However, the data accessible in the EudraVigilance EDSADRR database, for example, can be used for scientific research but show limited usefulness for the work of medical practitioners. For research purposes, scientific journals and publications remain as actual platforms to share the results of investigations, studies, observations, literature reviews, and meta-analyses. Publications on the neurological complications of vaccination are an example of the scale of the problem. These sources can sufficiently support the decision-making of healthcare professionals, especially those strengthened by the systematic approach, if healthcare professionals have the capacity and time for literature reviews. At the same time, social media represents a currently underutilized resource with high potential that could strengthen the pharmacovigilance system through active models if new technologies for big data analysis were appropriately applied. The involvement of social media in data mining could contribute to early warning systems for ADRs. In 2015, researchers proposed a methodology and prototype for filtering big data from social media to build an early warning system for ADRs, highlighting the ideas that data augmentation via partially supervised learning is effective in filtering ADR posts and that user-generated content in social media can provide timely information on ADR detection [68]. Later, researchers also proposed a framework to detect ADRs using internet user search data, in order that ADR events could be identified early, and the researchers successfully tested the method to reveal significant early warning signals for the side effects of medicines [69]. In addition, considering that underreporting has been identified as one of the limitations of the current pharmacovigilance system, social media could be an important tool for patient involvement in reporting.

The implications of this research are mainly related to the coverage of an understudied topic, while the limitations are devoted to an exploration of the passive pharmacovigilance system. A major strength of the present research is covering the gap of studies investigating the comparative magnitudes of prevalent medical conditions reported after COVID-19 vaccination, as well as their relevance to specific vaccine groups. The analyses were based on one of the largest datasets publicly available worldwide on vaccine-related adverse reactions. The limitations of the present study are connected with the fact that ICSRs do not represent conclusive evidence of a causal association between vaccine exposure and adverse reaction, as well as, data may be affected by under- or over-reporting bias due to public awareness of certain reactions. These limitations can be avoided by increasing the health literacy of society and smart engagement of patients in pharmacovigilance activities. Therefore, future research could be devoted to the study of the active pharmacovigilance model, which can sufficiently support healthcare professionals, patients, researchers, medicines manufacturers, and policymakers.

## 5. Conclusions

The pharmacovigilance system, including vaccine vigilance, was developed with the aim of improving the safety and effectiveness of drug use, and an imposing structure was created to achieve this aim. Nevertheless, several limitations in its functioning have been observed. Some of these are related to the classically dominant passive model of medicinal products’ post-marketing surveillance, which is based on the collection of reports on suspected ADRs, their long-term analyses, and causality assessments. Significant intellectual resources are involved in this evaluation process. However, the potential end users of the outcomes of this analysis—healthcare professionals and patients, who are initial reporters of ADRs—have ambiguously assessed the use of this system; therefore, underreporting has occurred and reduced the objectivity of relevant data. 

The development of an active model of pharmacovigilance that is strengthened through the use of social media data and the application of new big data processing methods could encourage the involvement of patients and healthcare professionals in the reporting of ADRs, accelerate the data acquisition process, and improve the effectiveness of using vigilance data for safer prevention and treatment process. Therefore, future research should focus on active pharmacovigilance models, as well as more in-depth research on vaccine-induced complications to reduce risks in the future.

The information obtained in this study on the predominantly reported medical conditions after COVID-19 vaccination and the relevant vaccine groups may be useful for health professionals, patients, researchers, and medicine manufacturers. Policymakers could benefit from reflecting on the design of an active pharmacovigilance model, making full use of modern information technologies, including big data analysis of social media for the detection of primary signals and building an early warning system.

## Figures and Tables

**Figure 1 vaccines-10-02115-f001:**
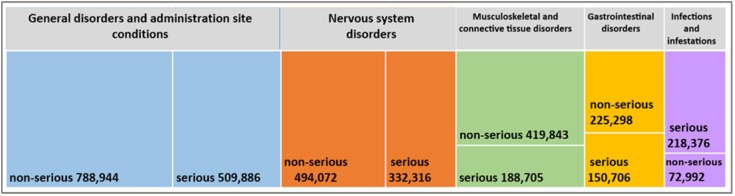
The number of Individual Case Safety Reports (ICSRs) on serious and non-serious suspected adverse reactions of COVID-19 vaccines authorized in the European Union, divided into groups of diagnoses largely represented by ICSRs in EudraVigilance European Database for Suspected Adverse Drug Reaction Reports (EDSADRR) (data on 17 September 2022, created by the authors based on statistical data [42]).

**Figure 2 vaccines-10-02115-f002:**
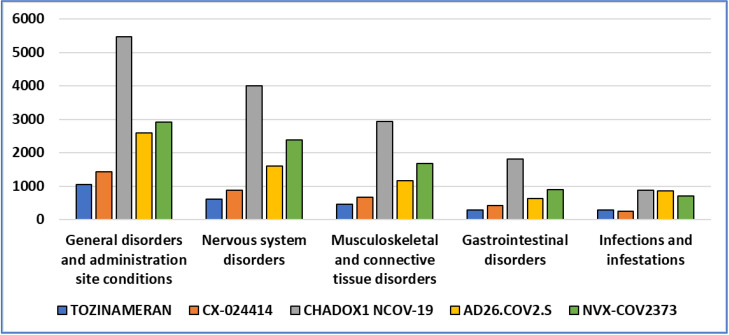
Relative frequency of Individual Case Safety Reports (ICSRs) on TOZINAMERAN, CX-024414, CHADOX1 NCOV-19, AD26.COV2.S, and NVX-COV2373 in EudraVigilance European Database for Suspected Adverse Drug Reaction Reports (EDSADRR) on five predominantly common medical condition groups, with data normalization using the total number of administered doses (number of ICSRs per 1 million administered doses) (data on 17 September 2022, created by the authors based on statistical data [42,43]).

**Figure 3 vaccines-10-02115-f003:**
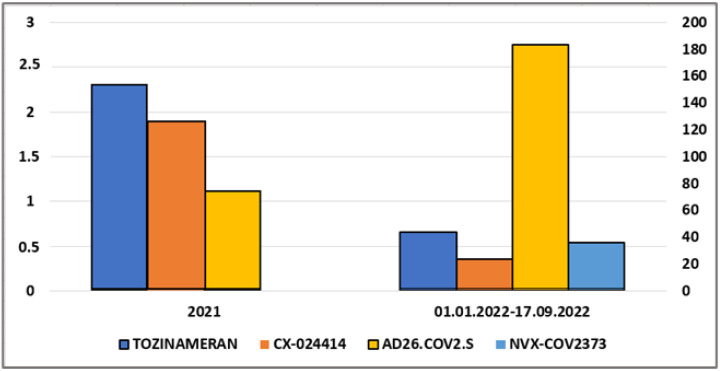
Relative frequency of Individual Case Safety Reports (ICSRs) for “chronic fatigue syndrome” on TOZINAMERAN, CX-024414, AD26.COV2.S, and NVX-COV2373 in EudraVigilance European Database for Suspected Adverse Drug Reaction Reports (EDSADRR), with data normalization using the total number of administered doses (number of ICSRs per 10 million administered doses) in 2021 (left scale) and in the period from 1 January 2022 to 17 September 2022 (right scale) (created by the authors based on statistical data [42,43]).

**Table 1 vaccines-10-02115-t001:** Studies selected from the PubMed database that used data on COVID-19 vaccines from the EudraVigilance database.

Authors and Publication Time	The Aim of the Study
Tobaiqy, Elkout, and MacLure (April 2021) [10]	The study aimed to identify and analyze the thrombotic adverse reactions associated with the Oxford-AstraZeneca vaccine.
Luo et al. (May 2021) [11]	The aim was to identify highly associated severe adverse events with Guillain-Barré syndrome (GBS) and develop prediction models for GBS.
Cari et al. (June 2021) [12]	The aim was to investigate: (i) Whether the frequency of severe adverse events is different in ChAdOx1 nCoV-19 COVID-19 (AstraZeneca) vaccine and BNT162b2 COVID-19 (Pfizer/BioNTech) vaccine recipients; (ii) whether the risk is limited to the adverse events described by the regulatory agencies; (iii) whether age and sex represent a risk factor; (iv) what is the risk in each age group.
Palladino et al. (June 2021) [13]	A quantitative benefit-risk analysis of ChAdOx1 nCoV-19 vaccine among people under 60 in Italy.
Douxfils et al. (July 2021) [14]	Hypotheses behind the very rare cases of thrombosis with thrombocytopenia syndrome after SARS-CoV-2 vaccination.
Gringeri et al. (September 2021) [15]	Evaluation of preliminary evidence on the safety profile of BNT162b2 (Comirnaty) from data analysis in EudraVigilance and adverse reaction reports from an Italian health facility.
Abbattista, Martinelli, and Peyvandi (October 2021) [16]	The aim was to assess the reporting rate of cerebral vein thrombosis as an adverse drug reaction (ADR) for the COVID-19 vaccines authorized in Europe.
Krzywicka et al. (November 2021) [17]	The analysis of cases of post-SARS-CoV-2-vaccination cerebral venous sinus thrombosis reported to the European Medicines Agency.
Tobaiqy et al. (November 2021) [18]	The study aimed to determine the frequency of reported thrombotic adverse events and clinical outcomes for three COVID-19 vaccines: Moderna, Pfizer, and Oxford-AstraZeneca.
Cari et al. (December 2021) [19]	An analysis of European data on cardiovascular, neurological, and pulmonary events following vaccination with the BNT162b2, ChAdOx1 nCoV-19, and Ad26.COV2.S vaccines.
Maltezou et al. (January 2022) [20]	Evaluation of anaphylaxis rates associated with COVID-19 vaccines are comparable to those of other vaccines.
van de Munckhof et al. (January 2022) [21]	The aim of this study was to evaluate whether the mortality of patients with cerebral venous sinus thrombosis (CVST) due to vaccine-induced immune thrombotic thrombocytopenia (VITT) after vaccination with adenoviral vector SARS-CoV-2 vaccines has decreased over time.
Ferner et al. (February 2022) [22]	The aim was to characterize the evolution over time of spontaneous reports of suspected ADRs to COVID-19 vaccines and to observe the effect of a publicized reaction (CVST) on reporting rates.
Montano (February 2022) [23]	The study aimed to provide a risk assessment of the adverse reactions related to the COVID-19 vaccines manufactured by AstraZeneca, Janssen, Moderna, and Pfizer-BioNTech, which have been in use in the EU and US between December 2020 and October 2021.
Krzywicka et al. (February 2022) [24]	The study aimed to assess the age-stratified risk of cerebral venous sinus thrombosis with and without thrombocytopenia after SARS-CoV-2 vaccination.
Yamashita, Takita, and Kami (March 2022) [25]	Extensive investigation to determine whether the number of death reports varied consistently over time after vaccination in the older population in Japan, the US, and European countries.
di Mauro et al. (May 2022) [26]	The study aimed to describe Individual Case Safety Reports (ICSRs) of impaired glucose metabolism events reported in the European database (EudraVigilance).
Lane, Yeomans, and Shakir (May 2022) [27]	The aim was to combine spontaneously reported data from multiple countries to estimate the reporting rate, and better understand risk factors for myocarditis and pericarditis following COVID-19 messenger RNA (mRNA) vaccines.
Luxi et al. (July 2022) [28]	The aim was to summarize the currently available evidence on frequency, risk factors, and underlying mechanisms of allergic reactions related to different COVID-19 vaccines, as well as on current recommendations for prevention and management of COVID-19 vaccine-allergic reactions, especially in those with a history of allergy.
Lane, Yeomans, and Shakir (July 2022) [29]	The study aimed to determine whether spontaneous reporting rates of myocarditis and pericarditis differed in immunocompromised patients compared with the whole population overall, and in terms of demographics, vaccine dose, and time-to-onset.
Cari et al. (September 2022) [30]	Evaluation of thrombotic events with or without thrombocytopenia in recipients of adenovirus-based COVID-19 vaccines.
Ruggiero et al. (2022 September) [31]	The study aimed to evaluate the capillary leak syndrome onset following receipt of COVID-19 mRNA vaccines (mRNA-1273 and BNT162b2) compared to viral vector vaccines (Ad26.CoV2-S and ChAdOx1-SARS-CoV-2).
García et al. (2022 September) [32]	The study aimed to analyze whether a disproportionate number of cases of subacute thyroiditis were reported in the EudraVigilance database for four COVID-19 vaccines (BNT162b2, mRNA-1273, ChAdOx1-S or Ad26.COV2.S).
Mascolo et al. (October 2022) [33]	The study aimed to investigate adverse events following immunization (AEFI) with COVID-19 vaccines during pregnancy.
Rodríguez-Ferreras et al. (November 2022) [34]	The study aimed to evaluate the relationship between Kikuchi-Fujimoto Disease and COVID-19 vaccination.
Hatziantoniou et al. (November 2022) [35]	Comparative assessment of myocarditis and pericarditis reporting rates related to mRNA COVID-19 vaccines in Europe and the US.
Oosterhuis et al. (November 2022) [36]	The study aimed to describe infrastructure, processes, and AEFIs reported for vaccine safety monitoring of COVID-19 vaccines during a large-scale vaccination campaign in the Netherlands.

**Table 2 vaccines-10-02115-t002:** COVID-19 vaccines included in the EudraVigilance database, doses administered in EU, and the percentage of adverse effects reported, until 17 September 2022 [42,43].

COVID-19 Vaccine	Doses Administered	Percentage of Adverse Effects Reported
mRNA vaccine Pfizer-BioNTech (TOZINAMERAN)	634.64 million doses	0.18
mRNA vaccine Moderna (CX-024414)	152.85 million doses	0.22
Vector vaccine AstraZeneca (CHADOX1 NCOV-19)	67.17 million doses	0.78
Vector vaccine Janssen (AD26.COV2.S)	18.67 million doses	0.37
Subunit vaccine Novavax (NVX-COV2373)	277,455.00 doses	0.49

## Data Availability

The data supporting reported results are available on the following websites: https://www.adrreports.eu/ (accessed on 17 September 2022); https://ourworldindata.org/covid-vaccinations (accessed on 17 September 2022); https://tools.meddra.org/wbb/ (accessed on 18 September 2022).

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
