# Peer review of "Vaccine Vigilance System: Considerations on the Effectiveness of Vigilance Data Use in COVID-19 Vaccination"

_vaccines, 2022, doi:10.3390/vaccines10122115_

Round 1

Reviewer 1 Report

The authors of the manuscript undertook to cover a very important topic, the topic of side effects from vaccines. From the text of the abstract, the reader can guess that the authors are trying to outline important problems associated with the accessibility and ease of use of the database in which such information is accumulated. However, the entire text of both the abstract and the rest of the manuscript poorly communicated. Authors use very long sentences, and the meaning of what they want to convey is lost. It is very difficult to understand. Long sentences with many thoughts and ideas merged are very difficult for comprehension. Finding or guessing the meaning of each sentence takes the reader a long time. Authors need to significantly improve the way they communicate their ideas and findings in their work. Otherwise, even reviewers have serious trouble reading the text. Poor communication can kill otherwise excellent research with important findings. In the case of this work, however, it is impossible to even discern the quality of the research and conclusions.

I suggest that the authors try to improve the way they communicate their findings and ideas. There are many web resources with good recommendations on how to make the information in the text more understandable. For example, the very first suggestion often found in these recommendations is to try to shorten sentences. The average sentence length in the abstract of the current version of the manuscript is 35, while the recommended gold standard is 25 or less.

https://insidegovuk.blog.gov.uk/2014/08/04/sentence-length-why-25-words-is-our-limit/

There are step-by-step instructions on how to improve communication clarity by adjusting sentence length:

https://www.enago.com/academy/how-to-optimize-sentence-length-in-academic-writing/#:~:text=Appropriate%20Sentence%20Length&text=Try%20to%20keep%20the%20average,a%20marathon%20or%20choppy%20sentences.

The web resources I linked to have many other helpful tips above also on how to improve idea communication for academic writers.

https://www.enago.com/academy/avoid-run-on-sentences-in-academic-writing/?utm_source=TrendMD&utm_medium=cpc&utm_campaign=Enago_Academy_TrendMD_0

I advise the author to study them and perhaps others to take the basic principles of good thought communication.

For those whose first language is not English, there are excellent free online translators that can significantly help improve scientific communication

https://www.deepl.com/translator

The real quality of this article can only be judged when the authors have significantly improved the language.

I would phrase the main question of this article as asking how accessible and user-friendly EudraVigilance 's database of vaccine side effects, with a particular focus on neurological complications, is for medical professionals and researchers. The authors also ask a more specific question: are there any other approaches for qualitative analysis of vaccine side effects related to neurological complications?

In my opinion, the topic chosen by the authors is very important, and there are few similar studies. There are currently 27 articles identified in Pubmed for the keywords “EudraVigilance database” and “COVID-19” and none of them analyze neurological complications. However, that's not because there aren't any. They are there, and several scientific papers have been published on the topic. I will name two: 1) Spectrum of neurological complications following COVID-19 vaccination, Neurol Sci. 2022; 43(1): 3–40. Published online 2021 Oct 31. doi: 10.1007/s10072-021-05662-9 2)  Safety of COVID-19 Vaccines: Spotlight on Neurological, Life (Basel). 2022 Aug 29;12(9):1338. doi: 10.3390/life12091338.  Complications

The question posed by the authors is a new one. No one of the researchers has ever raised the question the way they have raised it. As for publications on similar topics and related issues, I described them in my answer to the previous question.

As for what specific improvements should the authors consider regarding the methodology and what further controls should be considered, in answering the previous questions, I tried to justify that the authors' research topic might have value. However, I simply cannot answer the following questions. Because it is not possible to evaluate the research as such. The work is so poorly written that the quality of the research simply cannot be assessed. I cannot tell how the authors can improve the methodology, because I cannot even understand from their text how they applied it and what they got as a result.

Thus, summarizing the foregoing, I want to note that to evaluate the work, and especially how well it is done, how valuable the results, how well the authors have chosen the methodology, how well they illustrated the work and how well their conclusions can be only after the work will be completely rewritten from beginning to end and the authors will achieve clarity of presentation.

Author Response

Thank you for your valuable suggestions.

Reviewer 2 Report

The aim of the study was to investigate the availability of data on vaccination surveillance by healthcare professionals in the field of ADR. Vaccine safety data obtained using the detabase of the EudraVigilance, completed a review of the surveillance of side effects,  the Web of Science detabase was also used aim to search the literature on adverse effects reported after vaccinations for Covid-19. Pharmaceutical companies with a marketing autorization for distribution have been found to have limited control of ADR.The autors of the study sum up that it is necessary to implement a complementary model of supervision over safety of pharmacotherapy, especially ADR  logging after COVID-19 vaccines. Discussion written to the point using the available literature.

Author Response

Thank you very much for the inspiring valuation.

Reviewer 3 Report

Although this research is essential, however authors included social media, pharmacovigilance, and bibliometric assessments that lead to confusion. The manuscript is not clear. I suggest authors focus only on EudraVigilance and present the results. 

1. Abstract should be structured. 

2. Line 56 - 66: Is this the article's main focus? Then it should be presented at the end of the introduction. 

3. Since the study focused on EudraVigilance, I suggest removing unnecessary information from the background and describing - the EU system of COVID-19 PV activities. 

4. Social and mass media are not authentic sources and lack causality assessments. I recommend focusing only on formal PV activities that increase the readability and clarity of the work. 

5. Delete Lines 107 - 142. This created more confusion. 

6. Methods: Create sub-sections; it was complicated to read. 

7. Results are very descriptive and not clear to interpret. I suggest authors present them in a step-by-step manner according to the study objectives and differentiate the level of ADRs in EU-country and the type of vaccine. 

8. Line 381-422 is not scientific and create confusion in your work. I suggest deleting them. 

9. Discussion: Please split your discussion into four sections

A. Summary of findings

B. Comparison of findings with other studies

C. Implications of the work

D. limitations of the study

9. Conclusion: please describe the summary of your work followed by future directions to consider. 

10. The manuscript should be in line with the Vaccine journal guidelines; please make sure your manuscript has a 3000-3500 word limit. This helps to balance each section. 

11. Figure 6 is not clear, and 300dpi PNG pictures. 

Author Response

(The authors gave the same response as above.)

Round 2

Reviewer 1 Report

In my opinion, the goal of the authors of this article is to do a primary analysis of the data collected in the European Side Effects Data System and to make the results of the analysis available to the scientific community.

It is from this perspective that this paper and the data analysis conducted by the authors are of value to the scientific community, the medical community, vaccine manufacturers, and regulatory agencies. Interestingly, the authors have conducted a much broader analysis of the data than what is stated in the abstract of their paper.

The biggest drawback of the paper is that all the data obtained are presented as five pictures for different vaccines and these data are not normalized to each vaccine's share of the vaccine market in Europe. It is the comparative characteristics of the side effects of the different vaccines that are of great value.  I suggest that the authors normalize their data from pictures 1-5 by dividing the number of side effects from each category by the number of vaccines received by Europeans. This would give an approximate overall picture of how many side effects in each category occur, for example, for every 100,000 or million or 10 million doses of each type of vaccine received by the people of Europe. It seems to me that it would be worthwhile to make one general summarizing picture that would show, in a single coordinate system, the normalized number of side effects, that is, the number normalized for a certain number of doses of each type of vaccine administered. If I understand correctly, for this kind of summary illustration, the numerical data in Figures 1-5 should be divided by the data in Figure 7.

It would be good to show the different vaccines in this single illustration in separate categories, namely the vector vaccine category and the mRNA vaccine category and subunit vaccine category.

mRNA vaccine -Pfizer-BioNTech (TOZINAMERAN)  

mRNA vaccine -Moderna (CX-024414)

Vector vaccine - AstraZeneca (CHADOX1 NCOV-19)  

Vector vaccine -Janssen (AD26.COV2.S)

Subunit vaccine -Novavax (NVX-COV2373)

There are some less significant flaws in the paper as well. For example, the authors refer to post-acute sequelae of COVID-19 (PASC) as ME/CFS. These medical conditions or diseases do have much in common as symptoms. However, the medical and scientific literature distinguishes between these terms. PASC is a complication of COVID-19 – it is a proven fact. However, indirect association of ME/CFS with an infectious viral agent that provokes symptoms is suspected but not proven.

I suggest that the authors continue to work on improving the clarity of the presentation of their results. I highlighted one too long paragraph and suggested that it be broken into two or three, one thought, one paragraph. However, there are still too many long paragraphs in the article, where several thoughts are compressed into a single whole in such a way that they are difficult to comprehend and understand.

I would also like to comment on a few sentences in the text. “In this context, the authors of this article note their previous study on COVID-19 pandemic-revealed consistencies and inconsistencies in healthcare, which used statistical data to indicate that completely vaccinated persons tended to become COVID-19 infected more often than unvaccinated persons in the first four months of 2022 [34].”

The equalization of the number of COVID-19 cases among the unvaccinated and vaccinated in early 2022 is likely due to two reasons. The first is that most of the unvaccinated citizens were infected with the delta variant of the virus, which was spread in Europe in the second half of 2021 or in earlier infectious waves. As a result, they developed natural immunity. Therefore, the immune response from the natural disease became comparable to the immune defense from the vaccine. The second is the omicron virus variant broke through vaccine and natural immunity probably with equal frequency. This viral variant differed in antigenic composition from all previous variants, so both vaccine and natural immunity did not cope well with it.

In conclusion, I want to emphasize again that the authors have done, in my opinion, even more valuable work than they realize. However, for the value of this work to be well understood and conveyed to the readers of the article it is necessary to make small additional calculations, present them graphically and describe the obtained graphs in the results section.

I would also like to point out that the authors have a significant
chance to greatly increase the value of their work if they normalize
their data on side effects. In my opinion, it is interesting for the
scientific community to know not just how many side effects are reported
in the database, but how many were reported for every million doses of a
particular type of vaccine, for example. The authors have absolutely all
the data to do these calculations quickly. I am sure that if the
normalized data are presented, the citation rate of the article will
increase dramatically and the work will become of value to the
scientific community, the medical community, vaccine manufacturers, and
regulatory agencies. At the moment, the paper is in such a form that it
is unlikely to be cited.

Author Response

Thank you very much for the valuable suggestions.

Reviewer 3 Report

I have no other comments on authors work. 

Author Response

(The authors gave the same response as above.)
